# A²-TFG: ANALYTICAL AND ADAPTIVE TRAINING-FREE DIFFUSION GUIDANCE

## ABSTRACT

Training-free diffusion guidance uses an unconditional diffusion model and an off-the-shelf property predictor to generate samples with desired characteristics without further training. Typical predictors are any differentiable functions (*e.g.*, classifiers) that can evaluate the quality of generated samples. Existing works typically design the weights of the guidance term using heuristic rules combined with search-based tuning (e.g., the beam-search procedure used in TFG), resulting in fixed guidance weights for different samples. In this paper, we propose Analytical and Adaptive Training-Free Guidance (A²-TFG), which improves upon prior approaches in two aspects: **(1) Analytical**: we formulate an optimization objective that admits a closed-form solution for the guidance term weights under a locally linearized surrogate; **(2) Adaptive**: this closed-form solution varies with different inputs and timesteps rather than being fixed. Compared to heuristic rules or search-based tuning, these improvements lead to generally better performance in our experiments, while avoiding dataset-level hyperparameter search. We extensively validate the effectiveness of A²-TFG across six tasks combining three models such as Cat-DDPM, Stable Diffusion, and Audio-Diffusion with various task-specific targets, achieving superior performance over the vanilla TFG across most metrics.

## 1 INTRODUCTION

Over the years, various training-free guidance methods have been proposed, including DPS Chung & Ye (2022), FreeDoM Yu et al. (2023), and MPGD He et al. (2024). These methods have shown significant effectiveness across applications, providing robust solutions without extensive training data. The recently developed Training-Free Guidance (TFG) framework Ye et al. (2024) demonstrates that these methods, while effective in their respective domains, are special cases within a unified parameter space. By situating these approaches within a single framework, TFG enables a more comprehensive understanding

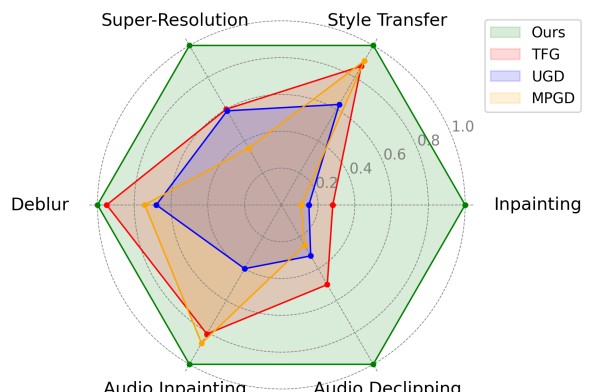

Figure 1: Quantitative comparison of A²-TFG and advanced models over six downstream training-free guidance tasks.

of their mechanisms. This unification lays the foundation for analyzing and improving training-free guidance methods in a principled way, and paves the way for novel techniques that leverage the strengths of existing methods, potentially enhancing performance and broadening applicability.

However, the current training-free guidance framework still relies on a non-trivial hyperparameter search to find per-task guidance, as shown in Figure 2(a). In TFG, this search is implemented via beam search on a validation subset, which substantially reduces but does not eliminate the additional

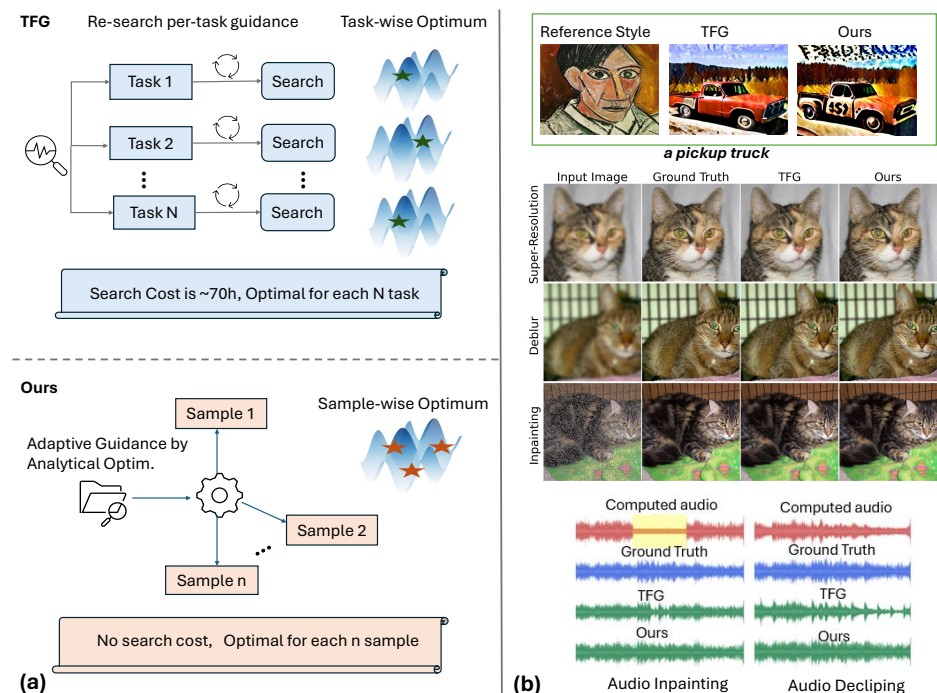

Figure 2: Key differences between TFG and our A$^2$-TFG: (a) TFG uses a validation-set search (implemented as beam search in the original paper) to select task-wise guidance parameters, incurring additional tuning cost; in contrast, our method analytically computes per-sample guidance weights at inference time, removing the need for a separate dataset-level tuning phase. (b) Qualitative results on style transfer, super-resolution, inpainting, and audio restoration show our approach yields outputs closer to ground truth than TFG in our experiments.

tuning cost. Such search procedures can yield slow convergence and high sensitivity to initialization and hyperparameters, and may lead to inconsistent outputs, as evidenced by the artifacts and deviations of TFG results in style transfer, super-resolution, inpainting and audio restoration, as shown in Figure 2(b). Moreover, in the absence of an analytical formulation for the guidance weights, it is difficult to reason about how the chosen values relate to the underlying objective or to provide guarantees beyond empirical performance.

Recent advancements in diffusion models have shown remarkable effectiveness across domains, including vision Ho et al. (2020a); Dhariwal & Nichol (2021a), audio Kong et al. (2020); Liu et al. (2023), molecular generation Hoogeboom et al. (2022); Xu et al. (2023), and 3D synthesis Luo & Hu (2021); Lyu et al. (2021), establishing them as foundational generative models at scale. Consequently, conditional generation—tailoring model outputs to meet user-defined criteria such as labels, attributes, and energies—has become a crucial downstream task. While conventional approaches like classifier guidance Ho & Salimans (2022) require specialized training for each conditioning signal, training-free guidance offers a more flexible alternative by leveraging off-the-shelf differentiable predictors without extra training overhead.

In this work, we reveal that parameter search in current training-free guidance frameworks can be interpreted as performing stochastic optimization over guidance parameters on a validation objective. To address these limitations, we introduce Adaptive Analytical Training-Free Guidance (A$^2$-TFG), which derives a closed-form, per-step update for guidance weights under a locally linearized surrogate objective within a unified training-free guidance framework. This analytical formulation bypasses the need for dataset-level hyperparameter search and provides insight into how the guidance weights relate to the underlying objective, while still being simple to implement. Through extensive experiments, we demonstrate that the analytical solution derived from A$^2$-TFG yields strong empirical performance in terms of both computational efficiency and generation quality, as showcased by the comprehensive comparisons and qualitative examples in Figure 1, even though it optimizes

a local surrogate rather than the final evaluation metrics directly. The contribution of this paper can be summarized as:

- **Theoretical Foundation**: We provide an analytical characterization of per-step guidance weights within a unified training-free guidance framework, deriving a closed-form solution under a locally linearized surrogate objective and clarifying the assumptions and limitations of this analysis.

- **Methodological Innovation**: We develop a novel optimization algorithm leveraging our analytical solution that eliminates the instability of stochastic optimization while significantly improving both computational efficiency and result consistency.

- **Empirical Validation**: We conduct comprehensive experiments demonstrating the practical advantages of our analytical approach in terms of generation quality, optimization stability, and computational efficiency across multiple benchmarks.

## 2 RELATED WORK

**Standard Guidance for Diffusion Models** Diffusion models Ho et al. (2020b); Song et al. (2021) generate high-quality samples via iterative denoising from Gaussian noise Sohl-Dickstein et al. (2015), capturing complex data structures. While early work focused on unconditional generation Ho et al. (2020b), a key advance is \*conditional generation\*—steering outputs towards desired criteria Dhariwal & Nichol (2021b); Saharia et al. (2022). **Training-based guidance** achieves this by incorporating explicit conditioning information during denoising, typically by training auxiliary models alongside the diffusion model. A prominent example is classifier guidance Dhariwal & Nichol (2021b); Nichol & Dhariwal (2021); Ho & Salimans (2022), which uses the gradient $\nabla_{\mathbf{x}_t} \log p_t(c \mid \mathbf{x}_t)$ from a time-dependent classifier to direct sampling towards condition $c$. While effective, this approach requires training specialized auxiliary models for each new condition, incurring high computational costs and limiting flexibility, thus motivating training-free alternatives.

**Training-free Guidance for Diffusion Models** To overcome the limitations of training-based approaches, various training-free guidance (TFG) methods have emerged, leveraging pre-trained models or differentiable objectives without task-specific retraining Hertz et al. (2022); Choi et al. (2021); Liu et al. (2021); Avrahami et al. (2022). Many rely on gradients from external predictors or energy functions, such as Diffusion Posterior Sampling (DPS) Choi et al. (2021); Liu et al. (2021), CLIP guidance Kim & Kim (2022), LGD Song et al. (2023), MPGD He et al. (2024), Free-DoM Yu et al. (2023), and UGD Bansal et al. (2023). The TFG framework Ye et al. (2024) unifies these gradient-based techniques. Alternative TFG paradigms include sampling-based methods (MC/SMC) Nisonoff et al. (2024); Cardoso et al. (2023); Venkatraman et al. (2024), which estimate posteriors and may reduce hyperparameter tuning, derivative-free approaches He et al. (2023); Li et al. (2024), alignment techniques (e.g., DPO Wallace et al. (2024)) that learn from preferences Kim et al. (2025); Xie & Gong (2024), and domain-specific methods Nisonoff et al. (2024); Rector-Brooks et al. (2024). Our proposed $A^2$-TFG operates within the gradient-based TFG paradigm and directly addresses the core challenge of optimizing guidance parameters.

**Optimizing Guidance Parameters in Diffusion Models** Optimizing guidance parameters is essential for effective diffusion guidance Nichol & Dhariwal (2021); Song et al. (2021); Kingma et al. (2021). In training-free, especially gradient-based, methods unified by frameworks like TFG Ye et al. (2024), selecting guidance strength (e.g., scaling factors) typically relies on hyperparameter search on a validation set (e.g., the beam-search procedure in TFG) Ye et al. (2024); Kingma et al. (2021); Song & Ermon (2020), which lacks adaptivity and theoretical grounding. Although such search is often run on a subset of the data to control cost, it still introduces an additional tuning phase whenever the task, dataset, or model changes. Although alternative paradigms (e.g., sampling-based Nisonoff et al. (2024); Cardoso et al. (2023), alignment-based Wallace et al. (2024)) and optimization techniques (e.g., adaptive sampling Nichol & Dhariwal (2021); Song et al. (2020)) exist, efficient and analytically derived parameter selection for gradient-based TFG remains under-explored. Our work, $A^2$-TFG, fills this gap by introducing a closed-form solution to replace costly search in gradient-based training-free guidance.

## 3 PRELIMINARY: UNIFIED TRAINING-FREE GUIDANCE

**Diffusion Guidance.** Suppose we have an unconditional diffusion model $\epsilon_\theta(\mathbf{x}_t, t)$, which essentially approximates the score function $\nabla_{\mathbf{x}_t} \log p_t(\mathbf{x}_t)$ learned from the training data. We aim to achieve conditional generation $\nabla_{\mathbf{x}_t} \log p_t(\mathbf{x}_t \mid y)$ through this model, where the condition could be a class label for category-specific generation, or partial observations of data for tasks like image/audio restoration. Using Bayes' theorem, we obtain

$$\nabla_{\mathbf{x}_t} \log p_t(\mathbf{x}_t \mid y) = \underbrace{\nabla_{\mathbf{x}_t} \log p_t(\mathbf{x}_t)}_{\text{Approximated by } \epsilon_\theta} + \underbrace{\nabla_{\mathbf{x}_t} \log p_t(y \mid \mathbf{x}_t)}_{\text{Guidance Term}}. \tag{1}$$

The second term on the right-hand side is the diffusion guidance, which steers the ODE / SDE trajectory toward the target conditional distribution. Intuitively, the guidance can be learned from data—for instance, when the condition is a class label, a classifier can be trained to model $p_t(y \mid \mathbf{x}_t)$.

**Training-Free Guidance (TFG).** At any timestep $t$ during denoising, we can obtain a prediction of the clean data $\mathbf{x}_0$, denoted as $\mathbf{x}_{0|t}$. If we assume $\mathbf{x}_{0|t}$ follows a distribution close to $\mathbf{x}_0$, we can leverage off-the-shelf property predictors to implement diffusion guidance. Such approaches are referred to as training-free diffusion guidance. Here, a property predictor can be any differentiable function $f_y(\mathbf{x}_{0|t})$, including classifiers, loss functions, energy functions, *etc*.

Existing TFG approaches, such as DPS Chung & Ye (2022), FreeDoM Yu et al. (2023), MPGD He et al. (2024), and UGD Bansal et al. (2023), construct the guidance term through heuristic rules by adding extra correction terms to the unified guidance formulation (see Eq. 1) together with more complex iterative procedures, demonstrating strong performance on their respective specialised tasks. However, their designs lack a unified theoretical basis and require careful hyperparameter selection. To this end, the Unified TFG Ye et al. (2024) is proposed. They find all previous TFG can be unified under a single formulation, with different methods only correspond to different hyperparameter selections. The hyperparameters space is defined as:

$$\mathcal{H}_{\text{TFG}} = \left\{ (N_{\text{recur}}, N_{\text{iter}}, \bar{\gamma}, \boldsymbol{\rho}, \boldsymbol{\mu}) : N_{\text{recur}} \in \mathbb{N}, \ N_{\text{iter}} \in \mathbb{N}, \ \bar{\gamma} \geq 0, \right.$$
$$\left. \boldsymbol{\rho}, \boldsymbol{\mu} \in (\mathbb{R}_+ \cup \{0\})^T \right\} \tag{2}$$

where $N_{\text{recur}}$ and $N_{\text{iter}}$ indicate steps of outer/inner iterations, $\bar{\gamma}$ a smoothing parameter. $\boldsymbol{\rho} = (\rho_1, \ldots, \rho_T)$ and $\boldsymbol{\mu} = (\mu_1, \ldots, \mu_T)$ are time-dependent scaling factors for two type of guidance terms. We skip discussion of the first three parameters as they can be easily determined, and set $N_{\text{recur}} = 1$, $N_{\text{iter}} = 1$ and $\bar{\gamma} = 0$ for simplicity [1]. As such, the Unified TFG writes as

$$\nabla_{\mathbf{x}_t} \log p_t(\mathbf{x}_t \mid y) \approx -\frac{\epsilon_\theta(\mathbf{x}_t, t)}{\sigma_t} + \rho_t \nabla_{\mathbf{x}_t} \log f_y(\mathbf{x}_{0|t})$$
$$+ \mu_t \nabla_{\mathbf{x}_{0|t}} \log f_y(\mathbf{x}_{0|t}). \tag{3}$$

where $\sigma_t$ is the noise schedule of the forward diffusion equation, satisfying $\mathbf{x}_t = \alpha_t \mathbf{x}_0 + \sigma_t \mathbf{z}$, where $\mathbf{z} \sim \mathcal{N}(0, \mathbf{I})$. The most important part is to determine the two sets of coefficients, $\boldsymbol{\rho}$ and $\boldsymbol{\mu}$. Unified TFG achieves this by searching over candidate values (implemented as beam search on a validation subset in the original paper) using the following objective:

$$\boldsymbol{\rho}^*, \boldsymbol{\mu}^* = \arg\min_{\boldsymbol{\rho}, \boldsymbol{\mu}} \mathbb{E}_{\tilde{\mathbf{x}}_0 \sim \text{val\_data}} \left[ \mathcal{L}_{\text{task}}(g_\Omega(\tilde{\mathbf{x}}_0), y) \right]. \tag{4}$$

Here, $g_\Omega(\tilde{\mathbf{x}}_0)$ is the sample generated by the diffusion model guided by the full parameter set $\Omega$ (which includes the specific $\boldsymbol{\rho}, \boldsymbol{\mu}$ being evaluated). The predictor $f_y$ is used within the generation process $g_\Omega$ to provide the guidance signal. $y$ denotes the target condition associated with $\tilde{\mathbf{x}}_0$. $\mathcal{L}_{\text{task}}$ is a suitable loss function (e.g., LPIPS for image restoration, FID for image quality) that measures the discrepancy between the generated sample $g_\Omega(\tilde{\mathbf{x}}_0)$ and the target condition $y$.

---

[1]For sake of easy understanding of our method, this is a little over-simplified, and we strongly recommend readers to read Ye et al. (2024) for a more comprehensive understanding of Unified TFG.

# 4 PROPOSED $\text{A}^2$-TFG

Searching for scaling factors $\boldsymbol{\rho}$ and $\boldsymbol{\mu}$ in the TFG framework has major drawbacks. The large search space ($2T$ parameters) leads to high computational cost. This method also produces fixed scaling factors ($\rho_t, \mu_t$) used at every timestep for all samples, restricting adaptivity. Grid search often finds sub-optimal solutions in complex, non-convex landscapes (see Figure 3). To overcome these issues, we propose $\text{A}^2$-TFG, an Analytical and Adaptive Training-Free Guidance method.

*Our key idea is to replace this global, discrete search with a local, analytical derivation of these scaling factors at each step $t$.* Specifically, $\text{A}^2$-TFG formulates a tractable optimization problem at each guidance step. The goal is to find the step-specific scaling factors, $\rho_t$ and $\mu_t$, that directly minimize the discrepancy between the current guided prediction and the target condition within a relevant feature space. This enables adaptive adjustment of guidance strength, moving beyond fixed parameters.

The other TFG parameters ($N_{\text{recur}}, N_{\text{iter}}, \bar{\gamma}$), which control broader loop structures and smoothing, are typically pre-set based on empirical findings or computational budget. Our analytical derivation focuses on $\rho_t$ and $\mu_t$, offering a more efficient and potentially more effective guidance mechanism. The complete procedure integrating these analytical updates is shown in Alg. 1.

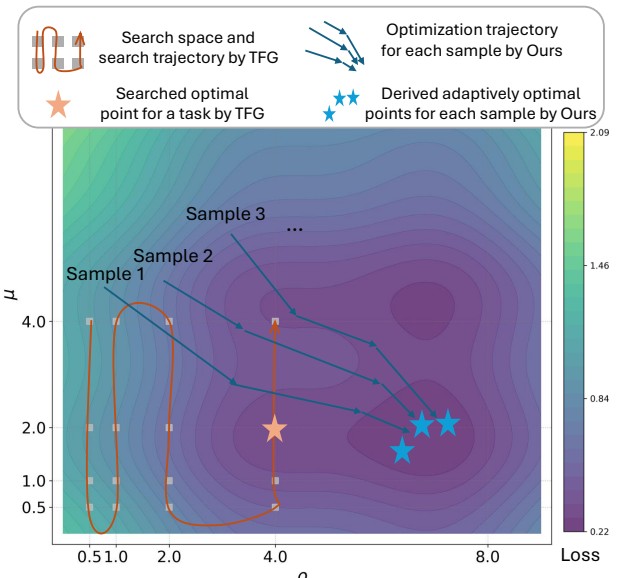

**Derivation of Analytical Updates** To implement this adaptive strategy for $\rho_t$ and $\mu_t$, we define a local optimization objective for each guidance step. The core idea is to determine the scaling factor (either $\rho_t$ or $\mu_t$, generically denoted as the scaling symbol $\zeta_t$ herein) that minimizes the discrepancy between the guided prediction and the target condition $y$ within a relevant feature space. To formalize the relationship, we employ a differentiable feature extractor $\mathcal{F}$ that encodes how the condition $y$ is derived from the ground truth $\mathbf{x}_0$, i.e., $y = \mathcal{F}(\mathbf{x}_0)$. Specifically, we focus on the predicted clean image $\mathbf{x}_{0|t}$, which can be obtained by the equation $\mathbf{x}_{0|t} = (\mathbf{x}_t - \sigma_t \epsilon_\theta(\mathbf{x}_t, t))/\alpha_t$. To distinguish, this prediction, which is obtained before the guidance update, is denoted as $\mathbf{x}_{0|t}^{\text{current}}$. The goal is to find the scale factor $\zeta_t$ that minimizes the following objective:

Figure 3: Optimization trajectories comparison between TFG and $\text{A}^2$-TFG in the $(\rho, \mu)$ space. *TFG (orange path)* denotes the validation-set search procedure over $(\rho, \mu)$ (e.g., beam search in the original implementation), which seeks a task-wise optimum but operates over a discrete grid of candidates. *Our $\text{A}^2$-TFG (blue trajectories for Samples 1–3)* analytically derives sample-specific optima (blue stars) under the local surrogate objective, enabling adaptive guidance without a separate dataset-level search phase.

$$\min_{\zeta_t} \|\mathcal{F}(\mathbf{x}_{0|t}^{\text{current}} - \zeta_t \cdot \Delta) - y\|^2, \tag{5}$$

where $\Delta$ is the corresponding guidance gradient, specifically $\Delta_t := \nabla_{\mathbf{x}_t} \log f_y(\mathbf{x}_{0|t})$ when solving for $\rho_t$, or $\Delta_0 := \nabla_{\mathbf{x}_{0|t}} \log f_y(\mathbf{x}_{0|t})$ when solving for $\mu_t$. This objective serves as a tractable proxy for improving alignment with the target condition at each step.

**Derivation of Adaptive Scaling Factors $\rho_t^*$ and $\mu_t^*$:** Assuming $\mathcal{F}$ is differentiable at $\mathbf{x}_{0|t}^{\text{current}}$ and locally linear, a first-order Taylor expansion gives:

$$\mathcal{F}(\mathbf{x}_{0|t}^{\text{current}} - \zeta_t \cdot \Delta) \approx \mathcal{F}(\mathbf{x}_{0|t}^{\text{current}}) - \zeta_t \cdot J_{\mathcal{F}}(\mathbf{x}_{0|t}^{\text{current}}) \cdot \text{vec}(\Delta), \tag{6}$$

where $J_{\mathcal{F}}(\mathbf{x}_{0|t}^{\text{current}})$ denotes the Jacobian matrix of $\mathcal{F}$ evaluated at $\mathbf{x}_{0|t}^{\text{current}}$. The term $J_{\mathcal{F}}(\mathbf{x}_{0|t}^{\text{current}}) \cdot \text{vec}(\Delta)$ is referred to as the *Jacobian-Vector Product (JVP)*. The *JVP* is implemented via automatic differentiation or traditional finite difference methods to reduce computational complexity. Substituting this into the objective transforms it into a linear least squares problem:

$$\min_{\zeta_t} \|\mathcal{F}(\mathbf{x}_{0|t}^{\text{current}}) - y - \zeta_t \cdot J_{\mathcal{F}}(\mathbf{x}_{0|t}^{\text{current}}) \cdot \text{vec}(\Delta)\|^2. \tag{7}$$

Taking the derivative with respect to $\zeta_t$ and setting it to zero gives the analytical solution for the optimal scaling factor:

$$\zeta_t^* = \frac{\langle \mathcal{F}(\mathbf{x}_{0|t}^{\text{current}}) - y, J_{\mathcal{F}}(\mathbf{x}_{0|t}^{\text{current}}) \cdot \text{vec}(\Delta) \rangle}{\|J_{\mathcal{F}}(\mathbf{x}_{0|t}^{\text{current}}) \cdot \text{vec}(\Delta)\|^2}. \tag{8}$$

This general solution is applied to derive both $\rho_t^*$ and $\mu_t^*$ as used in Algorithm 1.

Specifically, for $\rho_t^*$: Within each outer recursion step $r$, $\mathbf{x}_{0|t}^{\text{current}}$ is $\mathbf{x}_{0|t}^{(r)} = (\mathbf{x}_t^{(r)} - \sigma_t \epsilon_\theta(\mathbf{x}_t^{(r)}, t))/\alpha_t$, and $\Delta$ is $\Delta_t^{(r)} = \nabla_{\mathbf{x}_t^{(r)}} \log \tilde{f}(\mathbf{x}_{0|t}^{(r)})$. Substituting these into Eq. equation 8 yields the optimal $\rho_t^*$:

$$\rho_t^* = \frac{\langle \mathcal{F}(\mathbf{x}_{0|t}^{(r)}) - y, J_{\mathcal{F}}(\mathbf{x}_{0|t}^{(r)}) \cdot \text{vec}(\Delta_t^{(r)}) \rangle}{\|J_{\mathcal{F}}(\mathbf{x}_{0|t}^{(r)}) \cdot \text{vec}(\Delta_t^{(r)})\|^2}. \tag{9}$$

For $\mu_t^*$: Within each inner iteration $r'$, $\mathbf{x}_{0|t}^{\text{current}}$ is $\mathbf{x}_{0|t}^{(r')}$ (the prediction at the start of this inner iteration), and $\Delta$ is $\Delta_0^{(r')} = \nabla_{\mathbf{x}_{0|t}^{(r')}} \log \tilde{f}(\mathbf{x}_{0|t}^{(r')})$. Substituting these into Eq. equation 8 yields the optimal $\mu_t^*$:

$$\mu_t^* = \frac{\langle \mathcal{F}(\mathbf{x}_{0|t}^{(r')}) - y, J_{\mathcal{F}}(\mathbf{x}_{0|t}^{(r')}) \cdot \text{vec}(\Delta_0^{(r')}) \rangle}{\|J_{\mathcal{F}}(\mathbf{x}_{0|t}^{(r')}) \cdot \text{vec}(\Delta_0^{(r')})\|^2}. \tag{10}$$

These derived $\rho_t^*$ and $\mu_t^*$ provide adaptive, step-wise optimal scaling factors under the stated assumptions. By computing these values analytically at each step, our method replaces the need for a separate global hyperparameter search over $\rho$ and $\mu$, offering a more dynamic and efficient approach under the local surrogate objective.

**Discussion:** *(i) Justification of the Linear Approximation.* Our analytical step sizes in Eqs. 9 and 10 use a first-order Taylor approximation of $\mathcal{F}$. This is justified because optimization proceeds with small, iterative updates at each diffusion step, so $\mathcal{F}$ only needs to be locally linear in a small neighborhood—an assumption that generally holds. Moreover, the denominator $\|J_{\mathcal{F}} \cdot \text{vec}(\Delta)\|^2$ regularizes the step size, further ensuring the validity of linear approximation even for complex networks.

*(ii) Local Optimality and Convergence.* As reviewers noted, our method guarantees optimality only for the \*local, linearized\* objective at each step, not global or monotonic convergence. This trade-off yields a tractable, adaptive closed-form update. Empirically, these locally optimal steps effectively guide the process toward high-quality solutions, even if the objective does not always decrease monotonically.

*(iii) Independent vs. Joint Parameter Optimization.* Our framework optimizes scaling factors $\rho_t$ and $\mu_t$ independently for efficiency, as joint optimization would require expensive Hessian computations in high dimensions. This reduces the problem to two analytically solvable least-squares steps. Any suboptimality is mitigated by the iterative updates and the inner loop for $\mu_t$, which refines predictions within each timestep.

*(iv) Stability and Robustness.* Concerns about the stability of the Jacobian-vector product (JVP) and the accuracy of $\mathbf{x}_{0|t}$ are mitigated by the inherent noise $\sigma_t$ at each diffusion step, which regularizes the optimization and smooths gradients. Additionally, re-estimating $\mathbf{x}_{0|t}$ at every reverse step enables self-correction, limiting the propagation of early errors.

---

**Algorithm 1: Analytical and Adaptive Training-Free Guidance**

---

**Input:** Unconditional diffusion model $\epsilon_\theta$, Target predictor $f_\Omega$, Feature extractor $\mathcal{F}$, Steps $T$

Sample $\mathbf{x}_T \sim \mathcal{N}(\mathbf{0}, \mathbf{I})$

**for** $t = T, \cdots, 1$ **do**

   $\tilde{f}(\mathbf{x}) = \mathbb{E}_{\delta \sim \mathcal{N}(\mathbf{0}, \mathbf{I})} f(\mathbf{x} + \bar{\gamma}\sigma_t \delta)$ // Define measurement function, using Monte-Carlo method

   **for** $r = 1, \cdots, N_{recur}$ **do**

     $\mathbf{x}_{0|t} \leftarrow (\mathbf{x}_t - \sigma_t \epsilon_\theta(\mathbf{x}_t, t))/\alpha_t$ // Data prediction

     $\Delta_t = \nabla_{\mathbf{x}_t} \log \tilde{f}(\mathbf{x}_{0|t})$ // Gradient for $\rho_t^*$

       **Ours**: $\rho_t^* = \frac{\langle \mathcal{F}(\mathbf{x}_{0|t}) - y, J_\mathcal{F} \cdot vec(\Delta_t)\rangle}{\|J_\mathcal{F} \cdot vec(\Delta_t)\|^2}$ // Analytical solution

       **TFG**: $\rho_t^*$ // Grid search

     $\mathbf{x}_{0|t} \leftarrow \mathbf{x}_{0|t} - \rho_t^* \cdot \Delta_t$ // Apply $\rho_t^*$ update

     **for** $r' = 1, \cdots, N_{iter}$ **do**

       $\Delta_0 = \nabla_{\mathbf{x}_{0|t}} \log \tilde{f}(\mathbf{x}_{0|t})$ // Gradient for $\mu_t^*$

         **Ours**: $\mu_t^* = \frac{\langle \mathcal{F}(\mathbf{x}_{0|t}) - y, J_\mathcal{F} \cdot vec(\Delta_0)\rangle}{\|J_\mathcal{F} \cdot vec(\Delta_0)\|^2}$ // Analytical solution

         **TFG**: $\mu_t^*$ // Grid search

       $\mathbf{x}_{0|t} \leftarrow \mathbf{x}_{0|t} - \mu_t^* \cdot \Delta_0$ // Apply $\mu_t^*$ update

     **end for**

     $\mathbf{x}_{t-1} \leftarrow DDIM\_Sample(\mathbf{x}_{0|t}, \epsilon_\theta(\mathbf{x}_t, t))$

     $\mathbf{x}_t \sim \mathcal{N}(\alpha_t \mathbf{x}_{0|t}, \sigma_t \mathbf{I})$ // Recurrent strategy

   **end for**

**end for**

**Output:** Conditional sample $\tilde{\mathbf{x}}_0$

---

## 5 EXPERIMENTS

### 5.1 SETTINGS AND IMPLEMENTATIONS

**Task Settings** We design a diverse set of conditional generation tasks across vision, audio, and molecular domains. For vision, we evaluate image manipulation tasks such as Gaussian deblurring, inpainting, and super-resolution using Cat-DDPM Chen et al. (2023), and perform style transfer with Stable Diffusion Rombach et al. (2022). In audio, we consider declipping and inpainting using Audio-Diffusion Liu et al. (2023). Molecular generation experiments following the protocol of TFG Ye et al. (2024) are reported in the appendix. For each task, we evaluate two main metrics: guidance validity and generation fidelity. Guidance validity measures how well generated samples match target conditions, using metrics like classification accuracy for label guidance, LPIPS Zhang et al. (2018) for image reconstruction, Style score for style similarity, and DTW Müller (2007) for audio similarity. Generation fidelity assesses sample quality using FID Heusel et al. (2017) for images, CLIP score Radford et al. (2021) for content, and FAD Kilgour et al. (2018) for audio.

**Implementation.** We standardize diffusion timesteps ($T = 100$) and use the default DDIM sampling coefficients, and we set $N_{\text{recur}} = 1$, $N_{\text{iter}} = 4$ for all methods, following TFG Ye et al. (2024). In our framework, $\rho$ and $\mu$ use an *ascending configuration*, meaning that their schedules are monotonically non-decreasing over timesteps so that later steps use equal or stronger guidance than earlier ones, while scalars $\bar{\rho}$ and $\bar{\mu}$ are optimized via our search protocol. For baselines, we use original or published hyperparameters; if missing, we optimize scaling factors on $1/8$ of the data with up to 6 search iterations over $[0, 1000]$. In such cases, we follow the search procedures recommended in the original papers (e.g., beam search in TFG) to ensure fair comparison. Parameter tuning and comparisons use accuracy metrics, and final results report both accuracy and FID Heusel et al. (2017). Our method supports any differentiable predictor, including ResNet and ViT as in TFG.

### 5.2 COMPARISON WITH THE STATE-OF-THE-ARTS

**Main Results.** We evaluate our method against six baselines—DPS Chung & Ye (2022), LGD Song et al. (2023), MPGD He et al. (2024), FreeDoM Yu et al. (2023), UGD Bansal et al. (2023), and

Table 1: Comparison with state-of-the-art methods on vision restoration tasks. Metrics are averaged across settings. Best performance is in **bold**, second best is underlined .

| Methods | SR | | Deblur | | Inpainting | |
|---|---|---|---|---|---|---|
| | FID↓ | LPIPS↓ | FID↓ | LPIPS↓ | FID↓ | LPIPS↓ |
| DPS Chung & Ye (2022) | 91.80 | 0.4111 | 81.27 | 0.3761 | 98.45 | 0.4327 |
| LGD Song et al. (2023) | 59.54 | 0.2027 | 57.63 | 0.2219 | 43.49 | 0.0563 |
| MPGD He et al. (2024) | 63.51 | 0.2408 | 47.77 | 0.1436 | 37.72 | 0.0321 |
| FreeDoM Yu et al. (2023) | 82.34 | 0.3731 | 57.24 | 0.2198 | 39.83 | 0.0504 |
| UGD Bansal et al. (2023) | 50.54 | 0.1439 | 49.72 | 0.1569 | 38.02 | 0.0243 |
| TFG Ye et al. (2024) | 51.50 | 0.1429 | 42.57 | 0.1117 | **36.48** | 0.0129 |
| **Ours** | **39.83** | **0.0858** | **42.11** | **0.1067** | 36.78 | **0.0035** |

TFG Ye et al. (2024)—on image and audio manipulation tasks. As shown in Table 1 and Table 2, our method generally outperforms baselines across most metrics and tasks. On image restoration (Super-Resolution, Deblur, Inpainting), we achieve leading FID and LPIPS scores, indicating improved image quality and structure. For Style Transfer, our method is competitive, though LGD Song et al. (2023) excels in style adaptation. In audio tasks (Audio Inpainting, Audio Declipping), our approach achieves notably better DTW and FAD scores.

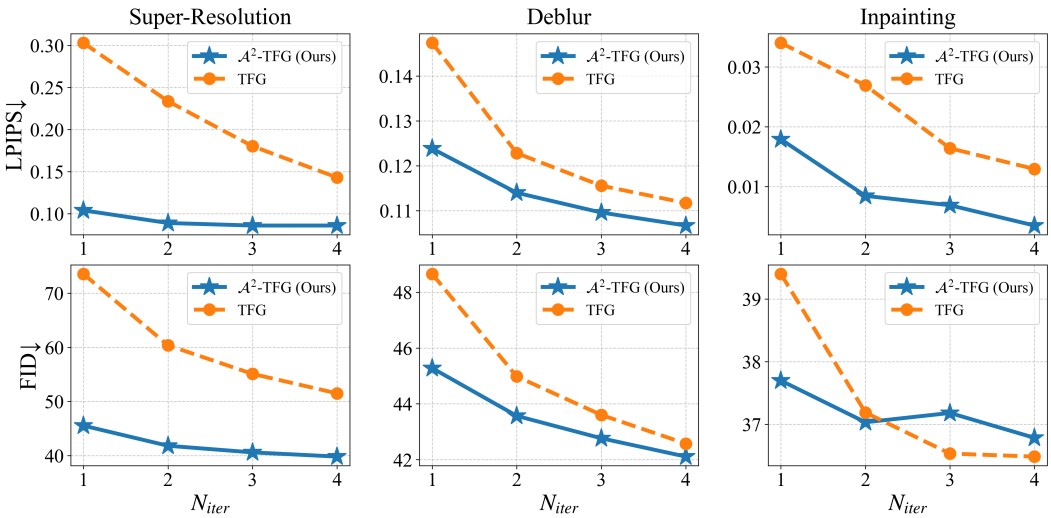

Figure 4: Comparison of LPIPS (top row) and FID (bottom row) scores between our method (A$^2$-TFG, solid blue line with stars) and TFG (dashed orange line with circles) over increasing iterations ($N_{iter}$) for Super-Resolution, Deblur, and Inpainting. Our method consistently achieves lower LPIPS and FID scores across all tasks and iteration counts.

Several key observations could be derived from the results. Different baselines work well on different tasks - LGD Song et al. (2023) performs best on style transfer, while MPGD He et al. (2024) works well on audio inpainting. This shows current methods often work well only on specific tasks. Our method performs well across all tasks, showing it can handle many different types of problems effectively. The improvement is most clear in complex tasks like audio manipulation, where our method reduces errors by up to 70% compared to other methods. These results show our method works well as a general solution that can handle many different tasks while maintaining good performance across different cases.

**In-depth Comparison with TFG.** We further analyze the performance difference between our method and TFG across different tasks and settings by examining how LPIPS and FID scores change with increasing iterations ($N_{iter}$). As shown in Figure 4, both methods show decreasing LPIPS values as iterations increase, indicating improved perceptual quality. However, our method achieves consistently lower LPIPS scores across all tested settings and tasks. This advantage is particularly noticeable in tasks like super-resolution (SR) and deblurring, where the performance gap remains

Table 2: Comparison with state-of-the-art methods on style transfer and audio manipulation tasks. Metrics are averaged across settings. Best performance is in **bold**, second best is underlined.

| Methods | Style Tr. | | Audio Inpaint. | | Audio Declip. | |
|---|---|---|---|---|---|---|
| | Style↓ | Clip↑ | DTW↓ | FAD↓ | DTW↓ | FAD↓ |
| DPS Chung & Ye (2022) | 56.36 | 31.39 | 32.67 | 3.002 | 44.68 | 3.689 |
| LGD Song et al. (2023) | 64.06 | **31.54** | 39.96 | 1.615 | 41.36 | 1.819 |
| MPGD He et al. (2024) | 37.38 | 31.05 | 4.503 | 0.1122 | 13.66 | 0.283 |
| FreeDoM Yu et al. (2023) | 52.56 | 31.45 | 34.99 | 1.571 | 13.22 | 0.557 |
| UGD Bansal et al. (2023) | 54.11 | 31.04 | 9.543 | 0.1612 | 11.10 | 0.220 |
| TFG Ye et al. (2024) | 38.81 | 29.60 | 4.818 | 0.0657 | 10.35 | 0.181 |
| **Ours** | **33.82** | 29.01 | **3.910** | **0.0647** | **3.46** | **0.071** |

stable or even widens with more iterations. The trend suggests that our method's improvements stem from fundamental algorithmic advantages rather than computational costs. When examining FID scores, a similar pattern emerges - our method maintains better performance across different settings while TFG shows higher sensitivity to parameter choices. This robust performance across varied conditions demonstrates the stability and generalizability of our approach compared to TFG's structure-dependent behavior.

### 5.3 FURTHER EMPIRICAL STUDY

**Quantitative Evaluation of Training-free Guidance.** As shown in Figure 4, we conduct comprehensive experiments comparing our method and TFG on three representative tasks: Super-Resolution, Deblurring, and Inpainting. The results demonstrate that our method consistently outperforms TFG in both perceptual quality, as measured by LPIPS, and generation quality, as measured by FID. In particular, for the Super-Resolution task, our method maintains significantly lower LPIPS scores (around 0.1) compared to TFG (above 0.15) across all tested iteration numbers, indicating a clear and stable advantage. Similar superior performance is observed in both Deblurring and Inpainting tasks, where our method achieves consistently lower metrics regardless of the number of iterations, further confirming its robustness and effectiveness in diverse scenarios.

**Distribution Analysis of Generation Quality.** To further validate effectiveness of our approach, we analyze the probability density distribution of LPIPS scores, as illustrated in Figure 5. The distribution analysis reveals that the samples generated by our method consistently concentrate at lower LPIPS values across all three tasks, which indicates better perceptual quality and more reliable results. This trend is especially prominent in the Super-Resolution task, where our method exhibits a clear peak at lower LPIPS values, and in the Inpainting task, where the distribution is more concentrated towards better scores. These compelling results substantiate that our method not only achieves better average performance but also maintains more stable generation quality, demonstrating its considerable advantage in both mean and variance of evaluation metrics.

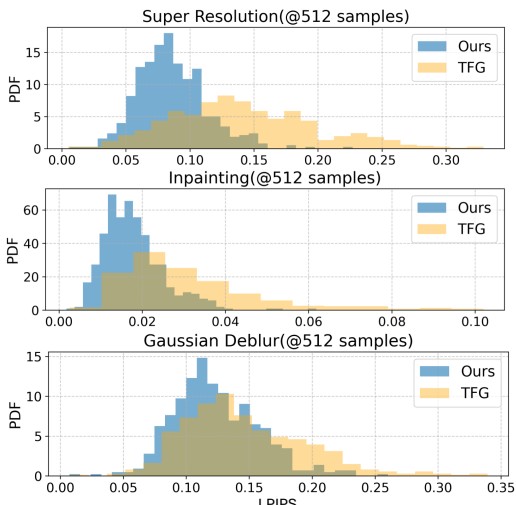

Figure 5: Probability density function (PDF) comparison between our method against TFG on SR, Gaussian Deblur, and Inpainting tasks with respect to LPIPS. It appears that most data points of ours concentrate at lower LPIPS values, indicating better performance.

**Computational Efficiency Analysis.** As shown in Table 3, we analyze computational cost, focusing on the trade-off between offline hyperparameter search and online inference. Methods like TFG require about 70 A100 GPU hours for hyperparameter tuning, which,

though a one-time offline process, often must be repeated in practice due to changes in data distribution or task. Parameters found on small subsets may not generalize well, resulting in costly re-tuning for each application or degradation type. In contrast, $A^2$-TFG eliminates this search phase entirely and only introduces minimal online JVP computation overhead during inference, increasing per-sample runtime from 5.45s to 5.59s.

The efficiency gains become more pronounced when considering the total cost of deploying TFG across multiple tasks or domains. While TFG's 70-hour search cost might seem acceptable for a single application, it scales linearly with the number of tasks, making our approach increasingly attractive for diverse applications.

Table 3: Computational cost comparison between TFG and our method. Ours eliminates hyperparameter search with minimal inference cost.

| Methods | Search Cost | Inference Cost |
|---|---|---|
| TFG Ye et al. (2024) | $\sim$70h | **5.45s** |
| **Ours** | **0h** | 5.59s |

## 6 CONCLUSION

We propose the first analytically-derived, unified training-free diffusion guidance, addressing instabilities in prior stochastic methods. Our method provides closed-form optimal guidance, removing costly hyperparameter searches. Experiments across domains show improved generation quality and lower computational cost over state-of-the-art baselines, while retaining robust, adaptive performance. The per-sample analytical adaptation generalizes to new tasks and modalities without extra training, highlighting the versatility and scalability of our framework.

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

## A  DECLARATION OF LLM USAGE

During the writing of the manuscript, we utilized a Large Language Model (ChatGPT) as a writing assistant. The scope of its usage was limited to **improving grammar, polishing sentences, and enhancing the clarity and fluency of this manuscript**. The method, claims, experimental results and conclusions are developed by the authors.

