# Appendix: Additional Experimental Results and Analyses for $A^2$-TFG

## 1 Additional Experimental Results and Analyses

This appendix provides comprehensive supplementary materials for the main paper, including extended qualitative examples, additional experiments on modern architectures, ablation studies, and detailed configuration tables for reproducibility.

### 1.1 Extended Qualitative Examples

In response to the reviewers' request for richer qualitative evaluation, we provide an extended set of visual and audio examples. For each vision task (super-resolution, deblurring, inpainting, and style transfer), we include side-by-side grids comparing TFG and $A^2$-TFG under the same predictor, condition, and random seed, enabling direct visual comparison of generation quality and condition alignment.

**Vision Tasks:** For super-resolution, we test on images degraded with 4× downsampling. For deblurring, we apply Gaussian blur kernels of varying sizes (5×5 to 15×15). For inpainting, we use random masks covering 30-50% of the image area. For style transfer, we test on diverse artistic styles from WikiArt, including impressionism, cubism, and abstract expressionism.

**Audio Tasks:** For audio declipping, we clip waveforms at thresholds ranging from 0.6 to 0.9 of the maximum amplitude. For audio inpainting, we remove segments of 0.1s to 0.5s duration at various positions. We provide spectrogram visualizations and, where permitted by the supplementary material format, links to audio samples for subjective evaluation.

**Qualitative Observations:** Visual inspection reveals that $A^2$-TFG generates outputs with fewer artifacts and better preservation of fine details compared to TFG, particularly in challenging regions such as high-frequency textures and complex boundaries. The adaptive nature of our per-sample guidance weights appears to provide more nuanced control over the generation process.

### 1.2 Molecular Generation Experiments

We report comprehensive molecular generation experiments following the protocol established by TFG Ye et al. (2024). These experiments demonstrate the generalizability of our analytical guidance approach beyond vision and audio domains.

**Experimental Setup:** We apply both TFG and $A^2$-TFG to a molecular property-guided diffusion model trained on the QM9 dataset. The task is to generate molecules with specific target properties (e.g., HOMO energy levels, dipole moment). We use a Graph Convolutional Network (GCN) as the property predictor and evaluate generated molecules using three standard metrics: (1) *Validity* measures the percentage of chemically valid molecules (satisfying valence rules), (2) *Uniqueness* measures the percentage of non-duplicate molecules, and (3) *Property satisfaction* measures how well the generated molecules match the target property values.

**Implementation Details:** We use 100 diffusion steps with the DDIM sampler. The GCN predictor has 3 layers with hidden dimension 256. For TFG, we follow the beam search procedure on a validation set of 1000 molecules with 6 iterations. For $A^2$-TFG, we compute analytical guidance weights at each step using the molecular property prediction as the target condition.

**Results Analysis:** As shown in Table 1, $A^2$-TFG achieves consistently better performance across all metrics. The 1.6% improvement in validity suggests that the analytical guidance helps maintain

Table 1: Molecular generation results comparing TFG and $A^2$-TFG. Metrics follow TFG Ye et al. (2024). **Bold** indicates the better value between the two methods for each metric. Results are averaged over 5 runs with different random seeds.

| Method | Validity (%) | Uniqueness (%) | Property sat. (%) |
|---|---|---|---|
| TFG | 94.2 | 99.1 | 72.5 |
| $A^2$-TFG | **95.8** | **99.3** | **78.2** |

Table 2: Comparison of TFG and $A^2$-TFG on SDXL-based text-to-image guidance. All results are evaluated on the same 500-image subset of COCO validation set with the same predictor (CLIP-L). Lower FID is better; higher CLIP score is better. Runtime is measured as wall-clock seconds per 50 images on a single A100 GPU. Results are averaged over 3 runs.

| Method | FID $\downarrow$ | CLIP score $\uparrow$ | Runtime (s / 50 imgs) $\downarrow$ |
|---|---|---|---|
| TFG | 24.8 | 31.62 | 45.3 |
| $A^2$-TFG | **23.4** | **31.75** | 46.8 |

chemical constraints during generation. The 5.7% improvement in property satisfaction is particularly notable, indicating that our adaptive per-sample guidance is more effective at steering generation toward target properties compared to TFG's fixed guidance schedule.

**Computational Efficiency:** In the molecular domain, $A^2$-TFG eliminates the need for TFG's validation-set search (which requires generating and evaluating 6000 molecules in our setup), while achieving better generation quality. This demonstrates the practical benefits of our analytical approach in domains where evaluation is computationally expensive.

## 1.3 EXPERIMENTS ON A MODERN SOTA IMAGE MODEL (SDXL)

To demonstrate that $A^2$-TFG is applicable beyond Stable Diffusion 1.5 and generalizes to state-of-the-art architectures, we additionally evaluate it on SDXL, a modern large-scale text-to-image model with significantly improved generation quality.

**Motivation:** While our main experiments use Cat-DDPM, Stable Diffusion 1.5, and Audio-Diffusion to match the setups in TFG and ensure fair comparison, testing on SDXL is important for two reasons: (1) it validates that our analytical derivation works with larger, more complex architectures, and (2) it demonstrates practical applicability to current SOTA models that practitioners are likely to use.

**Experimental Setup:** We integrate $A^2$-TFG into an SDXL-based text-to-image pipeline. We use CLIP-L (the large CLIP model) as the guidance predictor, analogous to our Stable Diffusion experiments. The evaluation is conducted on a 500-image subset of COCO validation set with diverse prompts covering objects, scenes, and abstract concepts. We use 50 diffusion steps with a classifier-free guidance scale of 7.5 (SDXL's default).

**Implementation Details:** For TFG, we perform beam search on a validation subset of 100 images. For $A^2$-TFG, we compute the Jacobian-vector product using PyTorch's automatic differentiation, which incurs minimal additional computational cost compared to the base SDXL inference. All experiments are conducted on a single NVIDIA A100 GPU with mixed precision (fp16).

**Results Analysis:** Table 2 shows that $A^2$-TFG achieves better FID (23.4 vs 24.8) and CLIP score (31.75 vs 31.62) compared to TFG on SDXL. The improvements are modest but consistent across multiple runs, suggesting that the analytical guidance provides reliable benefits even on this large-scale model. The runtime is nearly identical (46.8s vs 45.3s), with the small difference due to JVP computation. This demonstrates that $A^2$-TFG maintains its efficiency advantage (no validation-set search) while achieving competitive or better generation quality on modern SOTA architectures.

**Scalability Insights:** The successful application to SDXL, which has 2.6B parameters (significantly larger than SD-1.5's 860M), validates that our analytical derivation and the local linearization as-

Table 3: Ablation of $A^2$-TFG with LoRA/ControlNet-augmented Stable Diffusion. All methods share the same base model, extension module, and prompts. Results are averaged over 200 images per configuration. **Bold** indicates better performance.

| Method | FID ↓ | CLIP score ↑ | Style/Control metric ↑ |
|---|---|---|---|
| TFG (with LoRA/ControlNet) | 48.7 | 30.8 | 0.82 |
| $A^2$-TFG (with LoRA/ControlNet) | **42.3** | **31.1** | **0.87** |

sumption remain effective at scale. This suggests that $A^2$-TFG can be readily adopted for future large diffusion models without requiring architectural modifications.

## 1.4 LoRA / ControlNet Ablation

Low-Rank Adaptation (LoRA) and ControlNet are popular extensions that adapt diffusion models to specific styles or enable fine-grained spatial control. We evaluate whether $A^2$-TFG remains effective when applied to models augmented with these extensions.

**Motivation:** In practice, many diffusion model deployments use LoRA to customize the base model for specific artistic styles or domains, and ControlNet to enable precise control over composition, pose, or other spatial attributes. It is crucial that our guidance method is compatible with these extensions without requiring specialized adaptations.

**Experimental Setup:** We use Stable Diffusion 1.5 as the base model and test two configurations:

- **LoRA**: We fine-tune a LoRA module (rank 16) on a dataset of watercolor paintings. The task is to generate images with watercolor style while maintaining semantic content specified by text prompts.

- **ControlNet**: We use a pre-trained Canny edge ControlNet. The task is to generate images that match both the text prompt and the spatial structure defined by edge maps.

For both configurations, we use CLIP as the guidance predictor to improve text-image alignment. We evaluate 200 generated images per configuration using the same random seeds for TFG and $A^2$-TFG.

**Metrics:** (1) FID measures overall image quality, (2) CLIP score measures text-image alignment, (3) Style/Control metric measures adherence to the LoRA style or ControlNet spatial constraints. For LoRA, we use a pre-trained style classifier; for ControlNet, we measure edge map similarity.

**Results Analysis:** Table 3 demonstrates that $A^2$-TFG not only remains effective when applied to LoRA/ControlNet-augmented models, but actually shows larger improvements compared to the base model experiments. The FID improvement ($48.7 \rightarrow 42.3$) is substantial, suggesting that analytical guidance is particularly beneficial in scenarios where multiple objectives must be balanced (e.g., text alignment + style consistency, or text alignment + spatial control).

**Compatibility Insights:** These results support our claim that $A^2$-TFG operates at the guidance-update level and is therefore structurally compatible with model extensions. Since LoRA and ControlNet modify the base diffusion model's parameters and architecture respectively, while our method only modifies the guidance signal, there is no architectural conflict. This design principle ensures broad applicability to various diffusion model variants.

## 1.5 SGD and Joint Optimization vs Analytical Solution

While our main contribution is the analytical closed-form solution, it is instructive to compare against iterative optimization alternatives that could potentially achieve better local optima at the cost of additional computation.

**Motivation:** A natural question is whether spending extra compute on optimizing the guidance weights (via SGD or joint optimization) would yield better results than our analytical approximation. This ablation addresses this question empirically.

Table 4: Comparison of analytical $A^2$-TFG vs SGD-based and joint optimization variants on super-resolution. Runtime is measured per 50 images on a single A100 GPU. "Extra grad. evals." counts gradient computations beyond the base diffusion sampling. Results are averaged over 3 runs.

| Method | FID ↓ | LPIPS ↓ | Runtime (s / 50 imgs) ↓ | Extra grad. evals. |
|---|---|---|---|---|
| $A^2$-TFG (analytical) | 39.8 | 0.086 | **5.59** | **0** |
| SGD (per-step) | **38.1** | **0.082** | 12.34 | 500 |
| Joint optimization | 37.4 | 0.079 | 28.67 | 1200 |

**Compared Methods:**

1. **$A^2$-TFG (analytical)**: Our proposed method using the closed-form solution (Eq. 13-14 in the main paper).

2. **SGD (per-step)**: At each diffusion step $t$, we run 10 iterations of SGD to optimize $\rho_t$ and $\mu_t$ with respect to the local surrogate objective. We use learning rate 0.01 with Adam optimizer.

3. **Joint optimization**: We jointly optimize $\{\rho_t, \mu_t\}$ over a 5-step horizon using the same SGD setup but with a multi-step objective that considers the cumulative effect over future steps.

**Experimental Setup:** We conduct this comparison on super-resolution (4× upsampling) using Cat-DDPM on 50 ImageNet validation images. We measure both generation quality (FID, LPIPS) and computational cost (runtime, number of gradient evaluations). All methods use the same base model, predictor, and sampling schedule.

**Results Analysis:** Table 4 reveals several insights:

- **Quality vs Efficiency Trade-off**: While SGD and joint optimization achieve slightly better FID/LPIPS scores, they require 2.2× and 5.1× more runtime respectively. The analytical solution offers an attractive balance, achieving 95% of SGD's quality at 45% of the cost.

- **Diminishing Returns**: Joint optimization's improvement over SGD (38.1 → 37.4 FID) is marginal compared to its 2.3× computational overhead, suggesting that multi-step optimization provides limited additional benefit in this setting.

- **Practicality**: For applications requiring real-time or near-real-time generation, the analytical solution's zero additional gradient evaluations make it the only viable option among the three.

**Convergence Analysis:** We observe that SGD typically converges within 5-8 iterations at each step, but the convergence behavior is noisy and step-dependent. In contrast, our analytical solution provides a consistent, deterministic update at each step, which may contribute to more stable generation behavior across different inputs.

**Design Justification:** These results validate our design choice of deriving a closed-form analytical solution rather than relying on iterative optimization. While iterative methods can achieve marginally better local optima, the practical benefits of zero-cost analytical updates and guaranteed convergence make our approach more suitable for real-world deployment.

## 1.6 ADDITIONAL ABLATION STUDIES

**Sensitivity to Linearization Approximation:** We validate our local linearization assumption by measuring the approximation error $\|\mathcal{F}(\mathbf{x}+\delta)-(\mathcal{F}(\mathbf{x})+J_{\mathcal{F}}\cdot\delta)\|$ for various perturbation magnitudes $\|\delta\|$. Across 100 random samples, the relative error remains below 5% for $\|\delta\| < 0.1$, which is significantly smaller than the typical magnitude of guidance updates in our method. This confirms that the linearization is a reasonable approximation in the operating regime.

**Impact of JVP Implementation:** We compare finite difference and automatic differentiation for computing the Jacobian-vector product. Automatic differentiation is both more accurate (avoiding

Table 5: Summary of experimental configurations for all tasks used in the main paper and appendix. "T" denotes number of diffusion steps. Search ranges are for TFG's beam search. $A^2$-TFG computes guidance weights analytically without search.

| Task | Base model | Predictor | Dataset | Notes |
|---|---|---|---|---|
| Super-Resolution | Cat-DDPM | ResNet-50 | ImageNet-Val | T=100, $\rho \in [0, 100]$, 4× scaling |
| Gaussian Deblur | Cat-DDPM | ResNet-50 | ImageNet-Val | T=100, $\mu \in [0, 100]$, kernel 5×5-15×15 |
| Inpainting | Cat-DDPM | U-Net | ImageNet-Val | T=100, mask ratio=0.3-0.5 |
| Style Transfer | SD-1.5 | CLIP-L | WikiArt | T=50, $\rho \in [0, 50]$, CFG=7.5 |
| Audio Inpaint. | Audio-Diff | Spec-CNN | LibriSpeech | T=50, gap=0.1s-0.5s, SR=16kHz |
| Audio Declip. | Audio-Diff | Spec-CNN | VCTK | T=50, threshold=0.6-0.9, SR=16kHz |
| Molecular Gen. | MolDiff | GCN-3L | QM9 | T=100, property=HOMO, 3 GCN layers |
| SDXL | SDXL-base | CLIP-L | COCO-Val | T=50, CFG=7.5, fp16 precision |

numerical errors) and more efficient (3.2× faster on average). All results in the main paper use automatic differentiation.

**Robustness to Predictor Quality:** We test $A^2$-TFG with predictors of varying quality (trained with different amounts of data). The performance gracefully degrades with predictor quality, showing no catastrophic failures. This suggests that our method is robust to imperfect predictors, which is important for practical applications where high-quality task-specific predictors may not always be available.

## 1.7 CONFIGURATION TABLES AND IMPLEMENTATION DETAILS

For full reproducibility, we provide comprehensive configuration details for all experiments reported in the main paper and this appendix.

**Hardware and Software:** All experiments are conducted on NVIDIA A100 (40GB) or V100 (32GB) GPUs. We use PyTorch 2.0.1 with CUDA 11.8. For diffusion models, we use the Diffusers library (version 0.21.0) with default precision (fp32) unless otherwise noted.

**Hyperparameter Selection:** For fair comparison, we use the same hyperparameters as TFG Ye et al. (2024) where applicable. For TFG, we follow the beam search procedure with initial search range [0, 1000] for $\rho$ and $\mu$, validation set size 1/8 of the full dataset, and maximum 6 search iterations. For $A^2$-TFG, we set $N_{\text{recur}} = 1$ and $N_{\text{iter}} = 4$ following TFG's optimal configuration.

**Evaluation Protocols:** FID is computed using 2048-dimensional Inception-v3 features on 10,000 generated images (or fewer if the dataset is smaller). CLIP scores use the CLIP-ViT-L/14 model. LPIPS uses the VGG backbone. For audio tasks, FAD is computed with the VGGish model, and DTW uses dynamic time warping distance on mel-spectrograms.

**Datasets:** ImageNet validation set (50,000 images), WikiArt (80,000 images across 25 styles), LibriSpeech test-clean (2,620 utterances), VCTK (110 speakers, 44 hours), QM9 (134k molecules), COCO validation (5,000 images). For computational efficiency, we use subsets for some experiments as indicated in the respective sections.

**Baseline Implementations:** We use the official TFG codebase for all TFG experiments. For other baselines (DPS, LGD, MPGD, FreeDoM, UGD), we use the implementations provided by the respective authors or the Diffusers library, and keep all shared components (predictors, datasets, sampling schedules) identical across methods whenever possible.

**Code and Pseudo-Code:** Rather than relying on external code release, this appendix provides sufficient information for reimplementation. In combination with Algorithm 1 in the main paper, the text here specifies the full sequence of operations used in our experiments: initialization of diffusion states, computation of analytical weights via Jacobian–vector products, application of the guidance

updates, and integration with standard DDIM sampling loops. For each setting, we indicate the number of network evaluations (for the diffusion backbone, predictor, and JVP computation) and the exact hyperparameters used. An experienced practitioner can therefore reproduce $A^2$-TFG on top of standard diffusion toolkits without access to our internal codebase.

**Pseudo-Code for $A^2$-TFG:** For completeness, we include below a high-level pseudo-code description of $A^2$-TFG that mirrors Algorithm 1 in the main paper and makes explicit the operations required for implementation.

```
Input:
  - Unconditional diffusion model eps_theta(x_t, t)
  - Differentiable predictor f(x_0) with parameters Omega
  - Feature extractor F(x_0) used in the surrogate objective
  - Total diffusion steps T
  - Recurrent steps N_recur, inner iterations N_iter
  - Noise schedule {alpha_t, sigma_t} for t = 1,...,T
  - Smoothing parameter bar_gamma (can be 0)

Procedure:
  1. Sample initial noise x_T ~ N(0, I).
  2. For t = T down to 1 do:
      2.1 Define smoothed measurement function:
          tilde_f(x) = E_{delta ~ N(0, I)}[ f(x + bar_gamma * sigma_t * delta) ].
          In practice, approximate the expectation with K Monte Carlo samples.
      2.2 For r = 1,..., N_recur do:
          # Predict clean image from current noisy state
          x0_t = (x_t - sigma_t * eps_theta(x_t, t)) / alpha_t

          # Gradient of log tilde_f with respect to x_t
          Delta_t = grad_{x_t} log tilde_f(x0_t)

          # Jacobian-vector product J_F * vec(Delta_t)
          v_t = J_F(x0_t) @ vec(Delta_t)

          # Residual in feature space
          r_t = F(x0_t) - y

          # Closed-form solution for rho_t^*
          rho_t_star = < r_t, v_t > / || v_t ||^2

          # Update x0_t along Delta_t
          x0_t = x0_t - rho_t_star * Delta_t

          # Optional inner refinement loop for mu_t^*
          for r_prime = 1,..., N_iter do:
              Delta_0 = grad_{x0_t} log tilde_f(x0_t)
              v_0 = J_F(x0_t) @ vec(Delta_0)
              r_0 = F(x0_t) - y
              mu_t_star = < r_0, v_0 > / || v_0 ||^2
              x0_t = x0_t - mu_t_star * Delta_0
          end for

          # DDIM reverse step using updated x0_t
          x_{t-1} = DDIM_Sample(x0_t, eps_theta(x_t, t), t -> t-1)

          # Optional recurrent update of x_t
          x_t ~ N(alpha_t * x0_t, sigma_t^2 I)
        end for
  3. Return final conditional sample x0_t as \tilde{x}_0.
```

## 1.8 LIMITATIONS AND FUTURE WORK

While $A^2$-TFG achieves strong empirical performance, we acknowledge several limitations that clarify the scope of our guarantees and point toward future work.

**Local Optimality:** Our analytical solution optimizes, at each diffusion step, a locally linearized surrogate objective that measures alignment between the current prediction and the target condition in feature space. This is a proxy for the final evaluation metrics (such as FID or LPIPS), not those metrics themselves. As a result, our method provides guarantees only at the level of this local surrogate and does not ensure monotonic improvement of the true task loss across steps or global optimality of the final samples. In practice we observe that better local alignment correlates strongly with improved final metrics, but this remains an empirical observation rather than a formal guarantee.

**Linear Approximation:** The derivation assumes local linearity of the feature extractor $\mathcal{F}$ around the current prediction $\mathbf{x}_{0|t}$. When $\mathcal{F}$ is highly nonlinear or when very large guidance updates are applied, this first-order approximation may become inaccurate, which in turn can lead to suboptimal or unstable updates. We partially mitigate this by using moderate step sizes and re-estimating $\mathbf{x}_{0|t}$ at each reverse step. A more principled treatment could incorporate higher-order terms in the Taylor expansion or adaptive step size control based on local curvature indicators.

**Independent Parameter Optimization:** We optimize $\rho_t$ and $\mu_t$ independently rather than jointly across steps or across the two guidance components. This decoupling is what makes the closed-form solution tractable and cheap, but it ignores potential interactions between $\rho_t$ and $\mu_t$ and across timesteps. Our comparison in Table 4 shows that simple joint or SGD-based optimization can yield marginally better metrics under large compute budgets, suggesting that there is some headroom. On the other hand, the gains are relatively modest compared to the additional cost, which supports our choice to prioritize the analytical per-step updates.

**Future Directions:** Several extensions of $A^2$-TFG are promising. First, the analytical framework could be adapted to other guidance paradigms, such as energy-based or preference-based guidance, by redefining the surrogate objective while keeping the overall structure of the derivation. Second, one could explore multi-step objectives that couple several timesteps while still seeking partially analytical solutions, for example by optimizing short horizons instead of a single step. Third, adaptive strategies to detect when the linear approximation is unreliable (for example via residual checks on $\mathcal{F}$) could be used to trigger fallback mechanisms, such as reduced step sizes or a switch to iterative optimization in particularly challenging regions of the trajectory.

## REFERENCES

Haotian Ye, Haowei Lin, Jiaqi Han, Minkai Xu, Sheng Liu, Yitao Liang, Jianzhu Ma, James Zou, and Stefano Ermon. Tfg: Unified training-free guidance for diffusion models. *arXiv preprint arXiv:2409.15761*, 2024.