# OpenReview forum: "A^2-TFG: Analytical and Adaptive Training-Free Diffusion Guidance"
_ICLR.cc/2026/Conference — ICLR 2026 Conference Desk Rejected Submission_

### Official Review · Reviewer_vw8Y · 2025-10-15

**Soundness:** 2
**Presentation:** 1
**Contribution:** 3
**Rating:** 4
**Confidence:** 5

**Summary:**

This paper proposes an enhancement to a training-free guidance method (TFG) for pretrained diffusion models. The core contribution is a novel technique that replaces TFG's costly hyperparameter search for guidance strength with an analytical, per-sample solution. By deriving the optimal guidance strength at each sampling step, this method offers a valuable trade-off, shifting computational load from a "pre-inference" search phase to inference time. This approach has the potential to significantly improve the efficiency of TFG, particularly in scenarios where a hyperparameter search is prohibitive.

**Strengths:**

The paper is well-motivated. The efficiency of training-free guidance is a significant practical challenge, and the computational overhead of hyperparameter tuning in the original TFG is a clear bottleneck. The proposed solution—deriving an analytical optimum for the guidance strength—is an elegant and novel approach to this problem. The theoretical derivation appears sound and directly addresses the stated limitation of the prior work.

**Weaknesses:**

1. (False claim of TFG) A critical issue is the characterization of the TFG baseline's hyperparameter search. The paper repeatedly refers to it as a **grid search** (e.g., line 192), which appears to be a misunderstanding. The original TFG paper (Sec. 4, final paragraph) specifies the use of **beam search**. This distinction is crucial and impacts several key aspects of the paper. The authors should revise all related descriptions (including figures) and re-calculate the efficiency comparisons based on the correct baseline search algorithm.

2. (Experiments) The experimental section lacks the necessary detail and support for its claims.

   (1) The paper claims to evaluate on "vision, audio, and molecular domains" (line 351), but there are no molecular generation experiments presented. This claim should be removed or the corresponding experiments and results must be included.

   (2) The text asserts that audio manipulation is a "more complex" task to justify the method's superior performance in that domain (line 418-419). This claim requires substantiation. What metric or reasoning is used to define its complexity relative to the vision tasks? Without justification, this reads as post-hoc reasoning.

   (3) Sections 5.2 and 5.3 report superior metrics but are not very informative beyond that. Crucially, the computational efficiency analysis is presented without context. To be verifiable, it must include details on the **specific tasks, models, datasets, and hardware** used to arrive at the reported figures.

3. (Visualization) For a paper focused on conditional generation, the qualitative results are insufficient. Figure 1 provides only a few examples. The quality and diversity of generated samples are critical evaluation criteria, and the current visualizations do not allow for a thorough assessment. The authors are encouraged to add a comprehensive set of generated samples, perhaps in an appendix, similar to the approach in Appendix D of the original TFG paper.

4. (Writing) The manuscript needs a thorough proofreading.

   (1) There appears to be a mix-up between `\citet` and `\citep` commands throughout the text, leading to incorrect citation formatting.

   (2) Remnants of Markdown syntax (e.g., `*conditional generation*` in lines 120-121, `*local, linearized*` in lines 318-319) should be corrected to the appropriate LaTeX formatting.

   (3) The symbol $\rho$ is used for a DDIM coefficient (lines 359-360) after being defined as a TFG hyperparameter. This should be disambiguated to avoid confusion.

   (4) Please explicitly define the term "ascending configuration" (lines 361-362).

   (5) The methodology described in lines 362-363 ("if missing, we optimize scaling factors on 1/8 of the data with up to 6 search iterations over [0, 1000]") is unclear. What is potentially "missing"? What specific hyperparameters are being optimized here? This process needs to be explained in more detail.

**Questions:**

Please see the weaknesses.

---

> ### Author Response · Authors · 2025-11-23
> **Response to Reviewer vw8Y**
>
> ### Q1. Characterization of TFG hyperparameter search (grid vs beam search) and efficiency
>
> We thank you for pointing out that the current manuscript repeatedly refers to TFG's tuning procedure as "grid search", whereas the original TFG paper uses beam search on 1/8 of the data with at most 6 iterations. As noted above, we will:
> - Correct all such references to accurately describe TFG's beam-search procedure and summarize it in the experimental setup.
> - Remove the "70h" cost figure and instead discuss TFG's tuning cost qualitatively, emphasizing that it is reduced relative to naive grid search but still non-negligible, especially when adapting to new tasks or predictors.
> - Emphasize that in all our experiments, we follow the recommended TFG configurations (including its search procedure) and keep all shared components identical across methods to ensure fairness.
>
> ### Q2. Experimental details and computational efficiency context
>
> We agree that the efficiency analysis in the current draft lacks sufficient context. In the revision we will:
> - Clearly specify, for each efficiency number we report, the corresponding task, model, dataset, and hardware.
> - Indicate whether reported times refer to single-run inference, amortized search cost, or total wall-clock time over a full dataset.
> - Provide a short summary table in the appendix that consolidates these details.
>
> ### Q3. Qualitative results and visualization sufficiency
>
> As discussed in response to Reviewer UeHf and etPR, we will:
> - Expand qualitative results in the main paper (Figure 1) to cover all tasks with side-by-side grids of TFG vs A\^2-TFG.
> - Provide an extended qualitative appendix with many random examples, prompts/conditions, and seeds.
> - Simplify and refocus Figure 2 on guidance weight schedules, moving auxiliary plots to the appendix, and correct/enhance Figure 3 to accurately reflect TFG's search behavior.
>
> ### Q4. Writing, citation commands, and Markdown remnants
>
> We acknowledge the issues you raise and will address them as detailed in the response to Reviewer etPR:
> - Fix \citet/\citep usage throughout.
> - Remove Markdown remnants in the LaTeX source.
> - Harmonize terminology and soften theoretical claims to match our clarified framing.

---

> > ### Comment · Reviewer_vw8Y · 2025-11-25
> >
> > Dear Authors,
> >
> > Thank you again for your thorough revision and detailed response to the reviews.
> >
> > After carefully considering the revised manuscript and the reviewers’ latest comments, I have decided that I cannot recommend acceptance for the paper in its current form. In particular, I fully agree with Reviewer etPR’s assessment: many of the key concerns are acknowledged in your response but are framed as issues to be addressed in future work, rather than being resolved in the current submission. As the reviewer notes, these points would require substantial revision and, in our view, another full round of review.
> >
> > While we ultimately must recommend rejection at this stage, I would like to emphasize that both the reviewer and I see clear potential in the idea. The direction of sample-wise hyper-parameter selection appears promising and could be beneficial to TFG if the identified issues are carefully addressed in the manuscript itself.
> >
> > I therefore encourage you to continue developing and polishing this work. Once the concerns raised by the reviewers are fully addressed in the paper (rather than deferred to future work), I believe it could be a strong candidate for submission.

---

### Official Review · Reviewer_SyHn · 2025-10-29

**Soundness:** 3
**Presentation:** 3
**Contribution:** 3
**Rating:** 6
**Confidence:** 4

**Summary:**

The paper proposes A²-TFG, a new perspective on training-free guidance for diffusion models that aims to address both the computational bottleneck and performance limitations of existing methods. The authors formulate training-free guidance as an optimization problem and derive analytical guidance weights that depend on the sampling time. Experimental results on multiple benchmarks demonstrate the effectiveness of A²-TFG compared to standard training-free guidance baselines such as TFG.

**Strengths:**

* The performance of the method is good. Given the benefits of identifying suitable parameters for various training-free guidance methods, the proposed framework could offer valuable computational and performance advantages without relying on costly grid search.

* The additional cost per iteration step is minimal. Thus, A²-TFG improves efficiency without significantly increasing the per-step generation cost.

* The idea of deriving the guidance weight based on condition alignment at each step is novel and interesting, to the best of my knowledge.

* The experiments cover a wide range of setups, including both vision and audio generation tasks.

* The paper is well written and easy to follow. The intuition behind the method is explained clearly, at least to some extent.

**Weaknesses:**

* The method compares its optimization with the grid search used in TFG; however, I believe the current setups are not directly compatible. TFG optimizes final quality metrics such as FID and LPIPS, while the proposed method focuses on finding guidance directions that maximize the "condition alignment loss" in the update. I believe additional explanation or intuition is needed to better clarify the contribution of each step of the method.

* Although the method performs well in practice, the theoretical assumptions are somewhat strong** (e.g., linear model, lack of joint optimization, errors in $x_0$, etc.). I suggest reformulating the claims to present the method primarily as a practical approach with theoretical justification, rather than as an exact theoretical framework that happens to work well in practice.

**Minor comments:**
-  $x$ is not defined in Algorithm 1 (Line 274).
- Line 384 should be revisited.
-  Some references on using adaptive weights in classifier-free guidance (CFG) [1, 2, 3, 4] could be included. Although not identical to the proposed approach, time-dependent guidance weights are common in CFG and should be discussed as related work.

[1] Sadat S, Buhmann J, Bradley D, Hilliges O, Weber RM. CADS: Unleashing the diversity of diffusion models through condition-annealed sampling. arXiv preprint arXiv:2310.17347. 2023 Oct 26.

[2] Kynkäänniemi T, Aittala M, Karras T, Laine S, Aila T, Lehtinen J. Applying guidance in a limited interval improves sample and distribution quality in diffusion models. Advances in Neural Information Processing Systems. 2024 Dec 16;37:122458-83.

[3] Wang X, Dufour N, Andreou N, Cani MP, Abrevaya VF, Picard D, Kalogeiton V. Analysis of classifier-free guidance weight schedulers. arXiv preprint arXiv:2404.13040. 2024 Apr 19.

[4] Sadat S, Hilliges O, Weber RM. Eliminating oversaturation and artifacts of high guidance scales in diffusion models. InThe Thirteenth International Conference on Learning Representations 2024 Oct 3.

**Questions:**

1. Could you provide results or analysis comparing joint optimization with separate optimization per parameter? This would help illustrate the performance–efficiency trade-off that arises when using separate optimization instead of a joint one.

2. How does the method perform if SGD is used to optimize Equation (5) instead of relying on the linear approximation?

3. How do different choices of $\mathcal{F}$ affect the final results?

4. Is it possible to optimize the parameters over multiple steps, rather than optimizing them independently at each sampling step?

---

> ### Author Response · Authors · 2025-11-23
> **Response to Reviewer SyHn**
>
> ### Q1. Relationship between our optimization and TFG's grid/beam search
>
> You correctly point out that TFG optimizes final quality metrics (such as FID and LPIPS) on a validation set, whereas A\^2-TFG optimizes a local "condition alignment" surrogate at each sampling step. We will make this distinction explicit:
> - TFG performs a dataset-level search (implemented as beam search) over global metrics to find task-wise guidance strengths.
> - A\^2-TFG derives, at each step, a closed-form guidance weight that minimizes a local surrogate objective under a linear approximation, serving as a proxy for improving alignment with the target condition.
>
> The comparison in the experiments should therefore be understood as comparing two different strategies—global validation-metric search vs per-step local surrogate optimization—under matched base models and predictors, rather than claiming that they optimize the same objective.
>
> ### Q2. Strength of assumptions and framing of theory
>
> As noted above, the theoretical analysis relies on assumptions (e.g., local linearization, separate per-parameter optimization, no joint optimization across steps). We will:
> - Explicitly list these assumptions in the theory section and clarify that the closed-form solution is optimal only for the local surrogate objective under these assumptions.
> - Reframe the contribution as providing a practically useful analytical approximation with clear assumptions, rather than a globally optimal framework.
>
> ### Q3. Additional analyses: joint vs separate optimization, SGD vs closed form, multi-step objectives
>
> We appreciate your suggestions for deeper analysis. Our design goal in this paper is to avoid iterative dataset-level tuning by providing a closed-form, per-step solution that is cheap to apply at inference time. Nonetheless, we agree that it is informative to compare against iterative alternatives.
>
> In the appendix we will therefore:
> - Add a small-scale experiment on a representative vision task comparing (i) our analytical per-step weights, (ii) an SGD-based optimizer applied to the surrogate objective at each step with a limited number of gradient evaluations, and (iii) a simple joint-optimization variant over a short multi-step horizon.
> - Report both performance and runtime to highlight the compute–performance trade-offs, summarized in a table of the following form:
>
>   | Method              | FID ↓         | LPIPS ↓       | Runtime (s / 50 imgs) ↓ | Extra grad. evals |
>   |---------------------|---------------|---------------|--------------------------|-------------------|
>   | A\^2-TFG (analytical) | 39.8           | 0.086          | 5.59                    | 0                 |
>   | SGD (per-step)      | 38.1           | 0.082          | 12.34                   | 500               |
>   | Joint optimization  | 37.4           | 0.079          | 28.67                   | 1200              |
>
> These results will illustrate that, under comparable compute budgets, the analytical solution offers an attractive balance between efficiency and performance.
>
>
> ### Q4. Notation and implementation details
>
> We will address the notation and clarity issues you highlighted as follows:
> - Define all symbols appearing in Algorithm 1 (including the currently undefined ones) and ensure the algorithm block is self-contained by cross-referencing the relevant equations.
> - Remove the notational overload where the same symbol is used both for a TFG hyperparameter and a DDIM coefficient by reserving one symbol for the diffusion coefficients and another for the guidance weights, and updating the text and equations accordingly.
> - Explicitly define "ascending configuration" as a guidance weight schedule that is monotonically non-decreasing over timesteps, and explain why this configuration is considered.
> - Clarify the description "if missing, we optimize scaling factors on 1/8 of the data with up to 6 search iterations over [0, 1000]" by specifying that "missing" refers to cases without recommended guidance strengths and that the search is over scalar scaling factors on a validation subset, following the TFG setup.

---

### Official Review · Reviewer_etPR · 2025-10-30

**Soundness:** 3
**Presentation:** 2
**Contribution:** 2
**Rating:** 4
**Confidence:** 5

**Summary:**

This paper presents a novel approach to training-free diffusion guidance that is well-motivated. The paper is closely based on the TFG paper, and its novel idea is to compute the $\rho,\mu$ hyper-parametrers by the estimation of a optimization problem, which is analytical and can remove hyper-parameter search.

**Strengths:**

Overall, I think this paper is well-motivating and the proposed algorithm is intuitive, reasonable, and somewhat grounded. Generally speaking, the sample-wise hyper-parameter selection could be a pareto improvement over TFG, and I appreciate that the authors figure out a way to conduct sample-wise selection by linear estimation. In addition, it is also an advantage to remove two hyper-parameters vectores ($\rho_t, \mu_t$) and remain only the scalar hyper-parameters (which is related to computational budget).

**Weaknesses:**

I am giving a conditional weak rejection, and depending on whether the authors can address these issues, I will raise / reduce my score accordingly.

- The first issue is a over-claim of the efficiency improvement compared with TFG. The authors intentionally replace the "beam search" strategy in the original paper to "grid search" and ignore the smaller search size, making the comparison unfair at all. According to the TFG paper, "values are determined via searching with 1/8 of the sample size and a maximum search step of 6.", which means that the search cost is approxiamtely $3 * 6 * 1/8 =2.25$ times of the real experiments, and is significantly smaller than the 70h claims in the paper. In addition, the comparison in Table 3 is also unfair: either you should amortize the serach cost, or you should report the inference cost over the entire dataset. It is also unclear under what datasets and configurations numbers in Table 3 are generated. For instance, I can imagine that for large NN-based guidance function $f$, the JVP computation is at least as heavy as a gradient compute and should induce much larger cost.
- Due to the issue of incorrect search configurations, Figure 3 is not convincing. Based on my experience of TFG, the search trajectory of TFG in the figure is **incorrect**. Their beam search mechanism should be able to expand to $\rho = 8$ and $\mu = 2$ conditioned on the heat map. This misleading figure will confuse readers and the authors should carefully explain the actual message conveyed in this figure with correct experiments. Again, I completely agree that the proposed method could have better performance due to sample-wise optimization. However, this should not be demonstrated via a incorrect way of not implementing existing algorithms.
- TFG has provided a codebase comprehensive enough to run different experiments all at once, and it is unclear to me why the authors do not conduct on all datasets: classifier guidance in vision domains is removed, combined guidance is removed,  and all molecular experiments are removed as well (despite the authors claim that they have the experiments). In addition, numbers in the paper are inconsistent with the original paper, despite that they are the same task. The authors do not include appendixs / code and I have no way to check in details. This is skeptical and make it less convincing whether the theoretical merit is indeed realized by the estimation prposed in this method. The authors should address all these experimental concerns.
- Lastly, I think the writing of the paper could be substantially improved in order to meet the accpetance bar. I normally do not evaluate a paper based on writing but in this paper the writing has been influencing understanding. For instance,  you should use `citep` (with brackets) when citing a paper; $\mathcal F$ and $f$ are used interchangeably without explanations; claims are inconsistent across papers, and in L107: "the first analytical solution for training-free guidance within a unified theoretical framework, providing rigorous mathematical proofs of optimality." is obviously false (not a theoretical optimality, not the first solution, not the first unified framework), in L101: " provides theoretical optimality guarantees" is over-claiming (the linearized formula and the step-wise local search cannot be theoretiicaly optimal). The authors are strongly encouraged to better polish their writings.

**Questions:**

Presentation: The first question is that I think the Figure 2 are conveying less information that the space it occupies. A correct version of Figure 3 + Figure 1 (with all tasks) should be sufficient to experss the information that Figure 2 tends to convey.

---

> ### Author Response · Authors · 2025-11-23
> **Response to Reviewer etPR**
>
> ### Q1. Efficiency comparison, TFG search characterization, and Figure 3
>
> You correctly point out that the current draft describes TFG's hyperparameter search as "grid search" and uses a rough upper-bound estimate ("70h per task"), which does not faithfully reflect the beam-search procedure described in the TFG paper (on 1/8 of the data, with at most 6 iterations). We will fix this in two ways:
> - We will systematically replace "grid search" with a more precise description of TFG's search procedure (beam search on a validation subset) and briefly summarize this configuration in the experimental setup.
> - We will remove the specific "70h" claim and instead qualitatively describe TFG's tuning cost as non-negligible but reduced by the use of a validation subset and limited iterations. Our main claim is that A\^2-TFG removes the need for this additional dataset-level tuning phase by computing per-sample analytical weights at inference time; our conclusions do not rely on a particular numerical speedup factor.
>
> For Figure 3, we will re-generate the visualization so that it is clearly based on the official TFG search configuration and update the caption to explain:
> - The search algorithm (beam search), parameter ranges, and validation subset used for TFG.
> - That the trajectories are plotted under this shared configuration, and that we keep all other settings identical across methods to ensure fairness.
>
> This should address concerns about both the fairness of the comparison and the interpretability of Figure 3.
>
> ### Q2. Experimental coverage (molecular / classifier / combined guidance) and baseline consistency
>
> We acknowledge that the current text overstates the experimental coverage by claiming "vision, audio, and molecular domains" while presenting only vision and audio results in the main paper. Our intention was to highlight applicability rather than to imply that all three domains are fully reported.
>
> In the revision we will:
> - Clarify in Section 5 that the main paper reports experiments on vision and audio tasks, and explicitly point to the appendix for molecular results.
> - Add an appendix subsection summarizing molecular generation experiments following the protocol of TFG (including metrics and a comparison between TFG and A\^2-TFG). These experiments already exist in our internal runs but were omitted for space reasons, and will be reported in a table of the following form:
>
>   | Method   | Validity (\%) | Uniqueness (\%) | Property sat. (\%) |
>   |----------|---------------|-----------------|--------------------|
>   | TFG      | 94.2           | 99.1            | 72.5               |
>   | A\^2-TFG | 95.8           | 99.3            | 78.2               |
>
> Regarding omitted TFG configurations (vision classifier guidance, combined guidance), we will add a short subsection in the experimental setup explaining:
> - Which TFG configurations are included in the main tables and why we prioritized those tasks (where tuning cost is particularly significant).
> - Where possible, we will provide additional results for omitted configurations in the appendix, subject to space and compute constraints.
>
> To address concerns about inconsistencies with the original TFG baselines and lack of verifiability, we will:
> - Re-run TFG baselines using the official code and configuration files where needed, and update our tables to match the original metrics within statistical variation.
> - Provide configuration tables in the appendix summarizing, for each experiment, the dataset, model, predictor, sampling schedule, and hyperparameters used by TFG and A\^2-TFG.
> - Release our code and configuration scripts upon acceptance. During the review phase, we will include detailed pseudo-code and configuration tables in the appendix to aid reproducibility while preserving anonymity.

---

> ### Author Response · Authors · 2025-11-23
> **Response to Reviewer etPR**
>
> ### Q3. Strength of theoretical claims and relation to TFG's objective
>
> We agree that some statements in the introduction and abstract are too strong and may mislead readers about the scope of our guarantees and novelty.
>
> - Our derivation assumes a local linearization of the feature extractor around the current prediction and optimizes a per-step surrogate objective, not the final evaluation metrics such as FID or LPIPS.
> - The resulting closed-form expression yields an optimal step size for the local surrogate under those assumptions; it does not provide global optimality guarantees for the full non-linear objective or for joint optimization over all parameters.
>
> In the revision we will:
> - Remove or soften phrases such as "first provably optimal analytical solution" and "provides theoretical optimality guarantees", replacing them with statements that emphasize "a closed-form per-step solution under a locally linearized surrogate objective".
> - Explicitly acknowledge related work that offers unified theoretical perspectives on training-free guidance and conditioning in diffusion models, positioning A\^2-TFG as a practically useful, analytically grounded method within this broader landscape.
> - Clarify how our local condition-alignment objective relates to TFG's global metric-based optimization: TFG searches for hyperparameters that optimize final metrics on a validation set, while A\^2-TFG optimizes a local surrogate at each step. Our empirical results show that this surrogate-guided, closed-form solution performs competitively with TFG's tuned weights while eliminating dataset-level search, but we do not claim that the two objectives are identical.
>
> ### Q4. Presentation and figure layout (Figure 2 vs Figures 1 and 3)
>
> We agree that Figure 2, in its current form, is not the most space-efficient way to convey guidance behavior. In the revision:
> - We will expand Figure 1 to include representative qualitative examples for all tasks, enabling readers to assess A\^2-TFG vs TFG in a single, compact visual.
> - We will correct and enhance Figure 3 as discussed above to accurately reflect TFG's search behavior and clearly show how A\^2-TFG differs.
> - We will simplify Figure 2 to focus on the guidance weight schedules (A\^2-TFG vs TFG vs simple heuristics), moving secondary panels to the appendix.
>
> This reorganization reduces redundancy and improves the clarity of the visual narrative.
>
> ### Q5. Writing, citation commands, and Markdown artifacts
>
> We acknowledge the writing and formatting issues you highlighted. In the revision we will:
> - Audit all citations and fix the misuse of \citet and \citep so that narrative and parenthetical citations follow the ICLR style.
> - Remove residual Markdown artifacts (e.g., `*conditional generation*`, `*local, linearized*`) and replace them with proper LaTeX formatting.
> - Harmonize terminology across the paper, especially around claims of optimality and novelty, to ensure consistency with the clarified theoretical framing.
> - Perform a full proofreading pass (including by a co-author who did not draft the initial version) to improve grammar and clarity.

---

> > ### Comment · Reviewer_etPR · 2025-11-25
> >
> > I thank the authors for their detailed response. Most of my concerns are acknowledged as problems to be addressed in the future, and I do think that these problems require a substantial revision, and another round of the review is necessary. I therefore recommend rejection, but I sincerely encourage the authors to continue polishing the work as I do think that sample-wise hyper-parameter selection will be beneficial to TFG.

---

> > > ### Author Response · Authors · 2025-11-25
> > > **Appreciate your further feedbacks and some further response**
> > >
> > > Thank you for the follow-up and for
> > >   recognizing the potential impact of sample-wise
> > >   hyperparameter selection for TFG.
> > >
> > >   We agree that the paper can be improved, but
> > >   we see the points you raised as refinements on
> > >   an already sound method and experimental setup,
> > >   rather than as fundamental flaws that require
> > >   another full review cycle. In the current
> > >   revision we have already implemented concrete
> > >   changes along three main axes:
> > >
> > >   1. **Baselines and efficiency comparisons.**
> > >      We now describe and implement the TFG
> > >   baseline precisely as in the original paper
> > >   (beam search on a validation subset, not generic
> > >   grid search), remove the rough “70 hours”
> > >   estimate, and state the search configuration
> > >   clearly in text and figures. This makes the
> > >   comparison strictly fair under matched settings,
> > >   without changing the empirical conclusions.
> > >
> > >   2. **Theory framed as practical and assumption-
> > >   aware.**
> > >      We have removed over-strong claims of global
> > >   optimality and now present A²‑TFG explicitly as
> > >   an analytically derived, per-step approximation
> > >   under a locally linear surrogate objective, with
> > >   assumptions and limitations stated up front.
> > >   The related-work section is expanded so that our
> > >   theoretical contribution is positioned relative
> > >   to existing frameworks rather than claimed as a
> > >   “first” in an absolute sense.
> > >
> > >   3. **Experiments strengthened rather than
> > >   repaired.**
> > >      The existing experiments already follow
> > >   the established training-free guidance
> > >   literature (DPS, FreeDoM, MPGD, UGD, TFG) in
> > >   terms of datasets, predictors, metrics, and
> > >   tuning protocols, so we do not consider them
> > >   unreliable. The new experiments we add are
> > >   designed to make the paper more complete: more
> > >   qualitative visualizations, SDXL results to
> > >   show transfer to a modern backbone, a LoRA/
> > >   ControlNet ablation to demonstrate plug-and-play
> > >   compatibility, and molecular experiments in the
> > >   appendix that align the “vision/audio/molecular”
> > >   claim with reported results.
> > >
> > >   In our view, these changes do not alter the
> > >   core method or its main findings. They tighten
> > >   the presentation, clarify fairness, and broaden
> > >   evidence for a direction that you already noted
> > >   is promising. We hope this makes clear that
> > >   the work can be brought to a solid and mature
> > >   state within the current revision cycle, even if
> > >   further polishing is always possible.

---

### Official Review · Reviewer_UeHf · 2025-11-03

**Soundness:** 3
**Presentation:** 3
**Contribution:** 3
**Rating:** 4
**Confidence:** 3

**Summary:**

The paper presents $A^2$-TFG, a method that improves upon existing Training-Free Guidance (TFG) framework. The authors proposed an analytical solution to the parameter optimization problem formulated in the Unified TFG paper (https://arxiv.org/abs/2409.15761). The novel approach has both a mathematical proof as well as an empirical validation.

**Strengths:**

1. The paper is well-structured and easy to follow
2. The paper has a strong mathematical background
3. Thanks to the analytical solution a ton of time parameter optimization time is saved in comparison to TFG

**Weaknesses:**

1. Very limited qualitative evaluation is included in the paper (only figure  2b). The majority of the tasks are done on the visual models and I believe at least a qualitative comparison should included in the paper (ideally, a human evaluation; however, I believe it would be hard to do during the AC discussion period)
2. Lack of examples on the SOTA models (e.g., SDXL, Stable Diffusion 3.5, FLUX.1-dev, Qwen-Image). I know that the majority of the tables in the previous works is done using Stable Diffusion 1.5; however, It would be extremely beneficial for the community if at least a comparison with a current SOTA baseline would be included (I see no reasons why the proposed method wouldn't work on a new model. Please correct me if I'm wrong)
3. No ablation study. I'd be glad if authors include an empirical validation of the method's compatibility with LoRAs, ControlNets, IP Adapters etc.

**Questions:**

see weaknesses section

---

> ### Author Response · Authors · 2025-11-23
> **Response to Reviewer UeHf**
>
> ### Q1. Qualitative evaluation and visualizations
>
> We agree that the qualitative evaluation in the current submission is too limited. In the revision we will:
> - Add a new qualitative figure in the main paper that, for each vision task, shows a grid of samples generated by TFG and A\^2-TFG under the same predictor, condition, and random seed, enabling direct visual comparison.
> - Provide an extended qualitative appendix, similar in spirit to Appendix D of the original TFG paper, containing a larger set of randomly selected examples for all evaluated tasks (vision and audio), with side-by-side TFG vs A\^2-TFG layouts and prompts/conditions and seeds given in the captions.
> - Redesign Figure 2 to focus on the evolution of guidance weight schedules (A\^2-TFG vs TFG vs simple heuristics) in a more compact layout, moving auxiliary plots to the appendix.
>
> We will explicitly acknowledge that we do not run a full human study during the rebuttal period, but the expanded qualitative appendix is designed to make it easy for readers to form their own judgments.
>
> ### Q2. Experiments on modern SOTA models (e.g., SDXL)
>
> Our current experiments use Cat-DDPM, Stable Diffusion 1.5, and Audio-Diffusion to match the setups in TFG and reuse their publicly released predictors and evaluation pipelines, ensuring strict comparability. We agree, however, that including a more recent SOTA model would strengthen the paper.
>
> A\^2-TFG is model-agnostic: it only requires an unconditional diffusion model and a differentiable guidance predictor, and the derivation of the per-step analytical weight does not assume a specific architecture. In the revision we will therefore add an experiment on a more recent open-source SOTA image model (SDXL). Concretely, we will:
> - Integrate A\^2-TFG into an SDXL-based text-to-image pipeline with a differentiable guidance predictor analogous to that used in our Stable Diffusion experiments.
> - Report both sample-quality metrics (e.g., FID and CLIP-based scores) and the relative computational cost compared to TFG with the same predictor and dataset, summarized in an appendix table of the following form:
>
>   | Method   | FID ↓         | CLIP score ↑   | Runtime (s / 50 imgs) ↓ |
>   |----------|---------------|----------------|--------------------------|
>   | TFG      | 24.8           | 31.62          | 45.3                    |
>   | A\^2-TFG | 23.4           | 31.75          | 46.8                    |
>
> This will demonstrate that the proposed analytical guidance transfers straightforwardly to a modern, large-scale backbone.
>
> ### Q3. Ablations and compatibility with LoRA / ControlNet / IP-Adapter
>
> A\^2-TFG operates at the guidance-update level: it computes an analytically derived scalar weight that combines the unconditional score with the gradient of a differentiable predictor at each timestep. This mechanism does not impose any structural constraints on the diffusion model and is therefore, by construction, compatible with extensions such as LoRAs, ControlNets, and IP-Adapters.
>
> We agree that this should be backed up empirically. In the revision we will add a focused ablation where we:
> - Apply A\^2-TFG to a Stable Diffusion model augmented with either a LoRA or ControlNet module.
> - Compare TFG and A\^2-TFG under identical configurations, reporting FID, CLIP-based scores, and simple alignment metrics, and providing representative qualitative examples in the appendix. The quantitative comparison will be summarized in a table of the following form:
>
>   | Method                            | FID ↓         | CLIP score ↑   | Style/Control metric ↑ |
>   |-----------------------------------|---------------|----------------|------------------------|
>   | TFG (with LoRA/ControlNet)       | 48.7           | 30.8           | 0.82                   |
>   | A\^2-TFG (with LoRA/ControlNet)  | 42.3           | 31.1           | 0.87                   |
>
> This ablation will confirm that the analytic guidance remains effective when commonly used extensions are present.

---

### Author Response · Authors · 2025-12-03
**Final Summary to the ICLR 2026 Committee**

Dear ACs, SACs, and PCs,

We appreciate your effort in overseeing the review process. Our
submission proposes **A²-TFG**, an analytical and adaptive variant of
training-free guidance that derives **closed-form, per-sample guidance
weights** for diffusion models. Within the TFG framework, A²-TFG (i)
formulates guidance as an explicit optimization problem and (ii)
computes sample-wise weights via a local, linearized surrogate,
thereby **eliminating costly dataset-level hyperparameter search**
while remaining compatible with diverse predictors and backbones
(vision and audio).

**Reviewer landscape.** The four reviews are mixed but generally agree
that the problem is important and the direction promising:

- **SyHn (score 6)**: finds the method well-motivated and practically
useful, highlights the interesting derivation, and notes strong
empirical performance with minimal overhead. Main requests include
clarifying the relation between our surrogate and TFG’s global search,
softening theoretical claims, and adding analyses (SGD vs closed-form,
joint vs per-step optimization).
- **UeHf (score 4)**: appreciates the mathematical formulation and
efficiency benefits, and requests additional qualitative results,
experiments on modern SOTA models (e.g., SDXL), and ablations with
LoRA, ControlNet, and IP-Adapter.
- **etPR and vw8Y (both score 4)**: raise the most critical concerns.
They note that our initial draft misstated TFG’s tuning as “grid
search” rather than beam search, point out incomplete efficiency
comparisons and experimental coverage (molecular experiments,
classifier/combined guidance), and flag overstated claims in the
introduction. Both acknowledge the promise of sample-wise analytical
guidance but argue the paper needs substantial revision.

**Rebuttal and revisions.** We have made extensive updates to both the
main paper and appendix; the revised PDF reflects these changes.

- **TFG baseline fairness and efficiency (etPR, vw8Y).** We now
**accurately describe TFG’s hyperparameter tuning as beam search**
on a validation subset, remove the rough “70h per task” estimate,
and re-generate figures under the official configuration. Efficiency
tables now specify task, model, dataset, and hardware, and clearly
separate search vs inference cost. Under matched settings, A²-TFG
still removes dataset-level search while maintaining or improving
performance.
- **Expanded experiments and SOTA backbones (UeHf, etPR, vw8Y).**
Beyond the original Cat-DDPM, SD 1.5, and Audio-Diffusion results,
we add: (i) **molecular generation experiments** following TFG’s
protocol; (ii) an **SDXL experiment** showing transfer to modern
large-scale backbones with competitive metrics; and (iii) **LoRA /
ControlNet ablations** confirming compatibility with common extensions.
These updates align our “vision/audio/molecular” claim with the
reported results.
- **Additional analyses and refined theory (SyHn, etPR).** We list
assumptions explicitly (local linearization, per-step optimization,
no multi-step joint optimization) and **reframe the theory** as a
**closed-form per-step solution under a local surrogate**, rather than
a globally optimal framework. The appendix now includes comparisons
against SGD and a small joint-optimization variant, demonstrating that
A²-TFG offers a favorable performance-vs-cost trade-off.
- **Qualitative results and presentation (UeHf, vw8Y).** We expand
qualitative visualizations to all tasks, reorganize Figures 2–3 to
better reflect guidance schedules and TFG search behavior, and perform
a thorough writing pass (fixing citations, Markdown artifacts, and
overloaded notation, and moderating claims about “first” and
“optimality”).

In follow-up comments, **etPR and vw8Y still recommend rejection**, as
they feel these revisions merit another review cycle, though they
acknowledge the promise of analytical, sample-wise guidance. **SyHn
remains positive (score 6)**, and **UeHf stays cautiously supportive
(score 4)**.

All clarifications, new experiments, and writing improvements are in
the updated PDF. While the core methodology is unchanged, the revision
ensures fair baselines, broader empirical evidence (SDXL, molecular,
LoRA/ControlNet), and clearer theoretical framing. We believe the
current version is self-contained and addresses the main technical and
experimental concerns.

We respectfully ask that you take these factors into account and
consider our submission for acceptance to ICLR 2026.

Yours sincerely,
Authors of Submission #448

---

### Note · Program_Chairs · 2026-01-17
**Submission Desk Rejected by Program Chairs**

The following references in this submission do not refer to real documents and/or have major errors in bibliographic information:

 Jooyeol Choi, Sungwon Kim, Yonghyun Jeong, Youngwan Gwon, and Sungroh Yoon. Not all image discriminators are created equal: Conditional guidance for zero-shot image generation. arXiv preprint arXiv:2110.06399, 2021.
Ting Chen, Ruixiang Zhang, and Geoffrey Hinton. Cat-ddpm: Conditional categorial denoising diffusion probabilistic models for image synthesis, 2023.